# Plasmons in holographic graphene

Ulf Gran[1*], Marcus Tornsö[1] and Tobias Zingg[2,3]

**1** Department of Physics, Division of Subatomic, High Energy and Plasma Physics,
Chalmers University of Technology, SE-412 96 Göteborg, Sweden
**2** Nordita, Stockholm University and KTH Royal Institute of Technology,
Roslagstullsbacken 23, SE-106 91 Stockholm, Sweden
**3** Department of Physics and Helsinki Institute of Physics,
P.O.Box 64, FIN-00014 University of Helsinki, Finland

* ulf.gran@chalmers.se

## Abstract

We demonstrate how self-sourced collective modes – of which the plasmon is a prominent example due to its relevance in modern technological applications – are identified in strongly correlated systems described by holographic Maxwell theories. The characteristic $\omega \propto \sqrt{k}$ plasmon dispersion for 2D materials, such as graphene, naturally emerges from this formalism. We also demonstrate this by constructing the first holographic model containing this feature. This provides new insight into modeling such systems from a holographic point of view, bottom-up and top-down alike.
Beyond that, this method provides a general framework to compute the dynamical charge response of strange metals, which has recently become experimentally accessible due to the novel technique of momentum-resolved electron energy-loss spectroscopy (M-EELS). This framework therefore opens up the exciting possibility of testing holographic models for strange metals against actual experimental data.


# 1   Introduction

Holography is a powerful framework for computing the response functions of strongly cor-related matter, where, due to the absence of long-lived quasi-particles, perturbation theory is not applicable. When constructing holographic models one has to distinguish between *screened* response functions, which, for example, describe the response to changes in the internal, screened electric field $\mathcal{E}$ and the genuine, *physical* response functions [1] to the external electric displacement field $\mathcal{D}$. And it is the latter which would be the quantity that is actually being tuned directly in an experimental setup. The relation between screened and physical response is encoded in the dielectric function $\epsilon$, whose zeroes give self-sourced excitations in the material, one prominent example being the plasmon modes.

Thus, by analyzing the dielectric function $\epsilon$ we can both compare holographic models to experiments, as well as predict novel behavior of strongly correlated matter. Of particular interest is the little understood 'strange-metal' phase appearing e.g. above the critical temperature in high-temperature superconductors, and also in graphene in the form of the Dirac fluid. The dynamical charge response of strange metals, including plasmonic properties, has recently become experimentally accessible using the new method of momentum-resolved electron energy-loss spectroscopy (M-EELS) and an overdamped plasmon mode has been observed for small momenta [2]. Note that we use the term "plasmon" rather loosely to describe propagating self-sourced plasma oscillations, even in the absence of long lived modes. In recent work [3], we constructed the first holographic model of bulk plasmons, i.e. plasmons propagating in the interior of a three-dimensional material, and a key ingredient in the model was a very specific choice of boundary conditions. In this paper, we generalize these findings by giving a holographic prescription to model *all collective modes*, both in the longitudinal and transverse sectors. Furthermore, by extending the techniques developed in [3] to the situation of having charge carriers confined to a codimension one surface, we demonstrate how an $\omega \propto \sqrt{k}$ behavior generically emerges, thereby providing the first holographic model that accurately and naturally reproduces this characteristic feature of *2D plasmons*[1]. This behavior is not just a theoretical prediction, but has been confirmed in experiments with 2D layers, particularly graphene [4–6], within the accuracy that experiments allow. The reason why it is difficult to get highly accurate data for very low momenta is a phenomenon referred to as *wave localization*, where the wave-length for a plasmon at a specific frequency is significantly shorter than for light in vacuum at the same frequency. For graphene the difference can be up to a factor $\alpha^{-1} \sim 100$, where $\alpha$ is the fine structure constant [7]. This means that when trying to experimentally probe plasmon properties of a sheet of graphene the probing light and the sheet effectively decouple. A remedy employed is to add a grating to the graphene sheet, with a grating distance adjusted to the probe frequency. This provides an improved resolution,

---

[1]By this we mean a strictly 2D excitation, like in a sheet of graphene, in contrast to 'surface plasmons' which propagate in an interface between a metal and a dielectric material or air, where there is an exponential fall-off of the excitation in the direction transverse to the interface. We refrain from using the term 2DEG plasmons, i.e. plasmons in a 2D electron gas, since in the holographic framework there are in general no quasi-particles, and hence the relevant area of application is for systems without quasi-particle excitations like strange metals, e.g. the Dirac fluid in graphene.

but also requires a different physical sample for each probe frequency, making the systematic determination of the dispersion relation a very cumbersome and time-consuming process.

When considering linear response in holography, one generally obtains a set of coupled PDEs. Modes of the boundary theory correspond to specific solutions of these equations with a set of boundary conditions that would make the resulting boundary value problem over-determined. This is to be expected, however, since modes, by definition, correspond to particular intrinsic properties of the system. The conditions at the horizon are determined by regularity of the solution, and are thus inviolable. At the boundary, conditions are related to the holographic dictionary, with different choices corresponding to different responses one wishes to study. What has been well-established is that Dirichlet conditions give quasi-normal modes (QNMs), which are identified with poles of the holographic Green function. The knowledge of only these modes, however, is not sufficient to characterize all electromagnetic properties, as they just give the response to a screened electric field $\mathcal{E}$, which is 'internal' to the system. To obtain the physical response function, one needs to know the response to the, external, electric displacement field $\mathcal{D}$. In this work, we fill this gap and, based on that, draw conclusions on what conditions holographic models need to satisfy in order to describe all collective modes in the boundary theory.

This paper is structured as follows. In sec. 2 we will introduce the holographic notion of a collective mode and summarize how internal, physical, properties are related through the dielectric function. In sec. 3 we will then extend previous work [3] to illustrate how collective modes, both in the longitudinal and transverse sectors, correspond to specific choices of boundary conditions. This identification is consistent with the notion that these modes are, in condensed matter theory (CMT), usually identified with the vanishing of the dielectric function. Building up on this correspondence, we then continue in sec. 4 to lay out how these insights have to be incorporated into an effective holographic description of a strongly correlated codimension one system, like graphene, since the relative number of dimensions in which the charge carriers can move compared to the number of dimensions the potential permeates is an essential point. Based on these considerations, we argue in sec. 5 how a characteristic $\omega \propto \sqrt{k}$ dispersion for 2D plasmons naturally emerges when our formalism for identifying collective modes in holography is properly applied. We then corroborate this by constructing a holographic toy model that correctly reproduces this generic feature of the 2D plasmon dispersion relation. Furthermore, within the large parameter space of the holographic model, we will elaborate on the detailed agreement with results in regions where conventional condensed matter approaches are applicable.

## 2   Collective Modes

In this section we will introduce the holographic notion of a 'collective mode', i.e. a pole in the *physical* density-density response function[2] using the conventions and nomenclature of our previous work [3], which we review below to make the paper self-contained.

In a medium, the response to external electromagnetic fields is entirely described by Maxwell's equations,

$$d\mathcal{F} = 0, \quad d \star \mathcal{W} = \star \mathcal{J}_{ext}, \tag{1}$$

where, without loss of generality, we have absorbed the Maxwell coupling into the fields[3]. The

---

[2]See e.g. [1]

[3]The Maxwell coupling $e$ can easily be reinstated by re-scaling the 4-current $\mathcal{J} \to e\mathcal{J}$, remembering that e.g. $\chi_{sc}$ is defined to be quadratic in the charge density, c.f. (5), implying $\chi_{sc} \to e^2 \chi_{sc}$.

field strength $\mathcal{F}$ and induction tensor $\mathcal{W}$ are decomposed as,

$$\mathcal{F} = \mathcal{E} \wedge dt + \star^{-1}(\mathcal{B} \wedge dt), \tag{2}$$

$$\mathcal{W} = \mathcal{D} \wedge dt + \star^{-1}(\mathcal{H} \wedge dt). \tag{3}$$

Thus, $\mathcal{W}$ describes the 'external' electric displacement field $\mathcal{D}$ and the magnetic field strength $\mathcal{H}$, which are sourced by an external current $\mathcal{J}_{ext}$. In contrast, $\mathcal{F}$ describes the electric field strength $\mathcal{E}$ and magnetic flux density $\mathcal{B}$ inside the system, being the sums of the external fields and contributions from screening due to effects like polarization and magnetization in the material.

This distinction is important in order to decide which modes are to be considered 'collective modes', i.e. oscillations of the system in the absence of *external* fields [1]. The Green function gives the response to the induced current,

$$\mathcal{J} = (-\langle \rho \rangle \, dt + \boldsymbol{j}) = \star^{-1} d \star (\mathcal{F} - \mathcal{W}), \tag{4}$$

when the gauge field $\mathcal{A}$ is varied. This function encodes current-current correlations, and can be used to obtain the conductivity or the 'screened' density-density response,

$$\sigma_{ij} = -\frac{\langle \boldsymbol{j}_i \boldsymbol{j}_j \rangle}{i\omega}, \quad \chi_{sc} = \langle \rho \, \rho \rangle, \tag{5}$$

where $\sigma_{ij}$ is defined through Ohm's law, $\boldsymbol{j} = \sigma \cdot \mathcal{E}$. Due to functional identities of the Green function following from the continuity equation, it is also straightforward to derive elementary identities like,

$$\chi_{sc} = \frac{k^2}{i\omega} \sigma_L, \tag{6}$$

where $\sigma_L$ is the longitudinal conductivity. One has to keep in mind though, that these functions are 'constructs', as they describe the response to the screened fields $\mathcal{E}$ and $\mathcal{B}$, encoded in $\mathcal{F}$, and not the external fields $\mathcal{D}$ and $\mathcal{H}$ encoded in $\mathcal{W}$. For an electric response, the relation between the screened and unscreened response is encoded in the dielectric function

$$\epsilon_{ij} = \frac{\partial \mathcal{D}_i}{\partial \mathcal{E}_j}. \tag{7}$$

In particular, for the physical density-density response,

$$\chi = \frac{\chi_{sc}}{\epsilon_L}, \tag{8}$$

where $\epsilon_L$ is the longitudinal component of the dielectric function (7). Thus, a priori, collective modes, i.e. poles of the response function $\chi$, would be given by the poles of $\chi_{sc}$, as well as by the zeros of $\epsilon_L$. However, due to Maxwell's equations (1) there is a relation between the dielectric tensor and the conductivity,

$$\left(\varepsilon - 1 + \frac{\sigma}{i\omega}\right) \cdot \mathcal{E} = \frac{\boldsymbol{k} \times \mathcal{M}}{\omega}. \tag{9}$$

In isotropic media in particular, this implies that poles of $\sigma_L$, and hence of $\chi_{sc}$, are also poles of $\epsilon_L$, meaning that *it is only the zeros of the latter that characterize the poles of the physical response $\chi$*, which by definition correspond to collective modes as they represent propagating modes in the absence of external fields.

# 3 Holographic Boundary Conditions for Collective Modes

In this section we will extend previous work [3] to illustrate how both longitudinal and transverse collective modes correspond to specific choices of boundary conditions for perturbations. In the following we will use the term 'collective modes' to refer to poles of the physical response function $\chi$. That is, a function describing a response to an external, physical, source – in contrast to a response to the screened, internal, fields $\mathcal{E}$ and $\mathcal{B}$.

As argued in the last section, collective modes correspond to the zeros of the longitudinal dielectric function $\epsilon_L$. One type of collective modes is plasma oscillations inside the medium, so-called plasmons. It has recently been shown how to identify them holographically [3]. In that framework, these modes are again given by linear response in the electromagnetic sector, but one has to consider boundary conditions which are fundamentally different from Dirichlet conditions – as one would have when calculating QNMs, which would correspond to poles of the screened response $\chi_{sc}$. This is because collective modes characterize a situation with *self-sustained excitations*, i.e. vanishing external fields, $\mathcal{D} = 0$ and $\rho_{ext} = 0$, but non-vanishing fields in the interior of the material, $\mathcal{E} \neq 0$, giving rise to effects like propagating plasma oscillations. This makes the distinction to QNMs clear since imposing Dirichlet conditions on the gauge field leads, via the decomposition (3), to $\mathcal{E} = 0$, which according to Maxwell's equations (1) requires a specific external current to be applied in order to cancel the induced current in (4) so that Ohm's law is satisfied. This shows that QNMs are *driven excitations* from the viewpoint of the Maxwell theory on the boundary.

In the case of an isotropic system, there is only one transverse and one longitudinal counterpart of the dielectric function, denoted by $\epsilon_T$ and $\epsilon_L$, which provide all necessary information to identify the collective modes of the system. We proceed by deriving the boundary conditions for the corresponding bulk fields. Without loss of generality we impose $\delta\mathcal{A}_t \equiv \phi = 0$ and study a harmonic perturbation with momentum in the $x$-direction, and $y$ denoting any transverse direction. Then, after Fourier transforming, from Maxwell's equation (1) and the defining relation (4) in the absence of external fields it follows that

$$\omega^2 \delta\mathcal{A}_x + \delta\mathcal{J}_x = 0 \,, \tag{10}$$

$$\left(\omega^2 - k^2\right)\delta\mathcal{A}_y + \delta\mathcal{J}_y = 0 \,. \tag{11}$$

These conditions can be turned into boundary conditions for the bulk fields using the holographic dictionary, where we can relate the field strength on the boundary to the boundary value of the potential for the corresponding field in the bulk, as well as the current to the bulk induction tensor. Therefore, the explicit formulation is model-dependent, but in the class of Lagrangians usually studied in holography the current is generally related to the normal derivative of the corresponding potential at the boundary, such that,

$$\omega^2 \delta A_x + p_L \, \delta A'_x = 0 \,, \tag{12}$$

$$\left(\omega^2 - k^2\right)\delta A_y + p_T \, \delta A'_y = 0 \,. \tag{13}$$

These provide a unified description of all self-sourced solutions to holographic Maxwell theories. The functions $p_{L/T}$ are determined by the specific bulk theory at hand and the holographic dictionary [3]. They are generally bounded, but may depend on $\omega$ and $k$. In the case of a standard Maxwell action in the bulk, $p$ is constant. The type of mixed boundary conditions we arrive at are related to a double trace deformation in the QFT [3,8,9], corresponding to the random phase approximation (RPA) form of the Green function [10]. Therefore, our approach represents a natural extension of the conventional CMT techniques for computing the dielectric function to systems without long-lived quasi-particles.

It is also straightforward to reformulate the boundary conditions in terms of the dielectric function. This essentially follows from basic definitions and the relation (9), as well as noting

that in the absence of external fields we have $\mathcal{M} = \mathcal{B}$. Then, a bit of algebra reveals that (10) and (11) are nothing but

$$\epsilon_L(k, \omega) = 0, \quad \epsilon_T(k, \omega) = 0. \tag{14}$$

The first one, in accordance with the nomenclature in [3], we will refer to as the plasmon condition, but as shown above this condition yields *all* collective modes. The second one, i.e. the counterpart in the transverse direction, are transverse symmetric waves [1] involving transverse current fluctuations.

We emphasize that the collective modes corresponding to (12) and (13) are distinct from QNMs. The latter correspond to poles of the Green function for internal correlators, while the former yield collective modes that can be directly accessed via experiments. Thus, identifying all collective modes is a crucial step to relate holographic results to actual data and, for the holographic Maxwell theories we consider, all collective modes correspond to (12) and (13), as explained above.

## 4 Holographic Graphene as Codimension One Boundary Theory

All electromagnetic phenomena can be described through the field strength $\mathcal{F}$ and the induction tensor $\mathcal{W}$. While it has been known for a while how to identify the former in the boundary theory, the correspondence for the latter seems to have gone unmentioned until recently [3]. In the following we will examine consequences of this identification with regard to codimension one systems, like graphene. For a different holographic bottom-up model of graphene see [11].

Holography has been established as an effective description of strongly correlated systems. However, while the strength of interaction is large in these systems, one does not wish to change the fundamental nature of the interaction, respectively test charges, itself. And the latter is intricately connected with the dimension in which particles interact, as creation and annihilation operators are determined as distribution-valued operators for point-like sources. This becomes relevant in systems like graphene, certain high-$T_c$ superconductors and other compound materials, where charge carriers move in layers – or even edges, in some cases. I.e. positioning is confined to a lower number of dimensions, but the charge carriers are, in essence, still subject to interactions determined by the '$1/r$' Coulomb potential in three spatial dimensions – in contrast to, e.g., the '$\ln r$' potential in two dimensions. In a proper description of such a material, this feature must therefore also be incorporated into an effective model. Meaning that while the system one wishes to study is effectively $(2 + 1)$-dimensional, one has still to keep in mind that it is composed of particles whose electromagnetic interaction is determined by Maxwell's equations in $(3 + 1)$-dimensions.

From a holographic perspective, this requires to incorporate a mechanism that keeps particles 'in place'. In a top-down construction, this is usually achieved via embedding D-branes, which is also how aspects of graphene and related systems were holographically modeled previously [12–17], and is illustrated in figure 1. This description, however, will create some difficulties from a practical standpoint. As pointed out in previous work [3], gravitational back-reaction seems crucial to determine material properties like plasmon excitations. For D-branes, however, going beyond a probe limit – in which back-reaction is neglected – is a rather difficult task where, in general, no efficient computational methods are at hand.

Nevertheless, we will illustrate in the following how the interplay of confinement of charge carriers to a lower-dimensional subspace and gravitational back-reaction are the basic features that make a holographic description provide physically realistic results. The important point to keep in mind is that it is not the *details* of the gravitational interaction with the brane that is

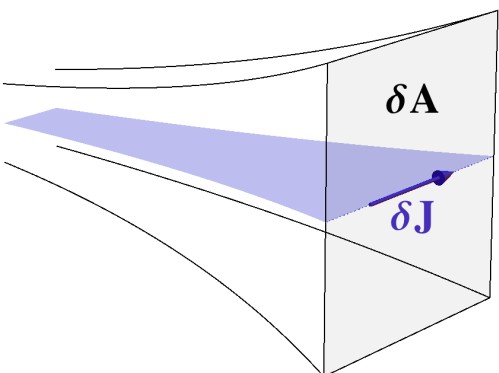

Figure 1: Schematics of a holographic setup for a physically realistic model for graphene. While the induced current $\delta J$ on the boundary is confined to $2+1$ dimensions, the boundary potential $\delta A$ lives in $3+1$ dimensions. Therefore, a realistic holographic setup would require a brane-construction that keeps particles in place and electromagnetic phenomena are described by the bulk physics projected onto the brane.

relevant, just that there is *some* mediating interaction with the background. Thus, as a proxy, consider the gravitational interaction 'projected' onto the brane – which is sufficient, since we are only interested in studying phenomena due to particles that are, ultimately, constrained to not being able to move beyond the dimensions in which the brane extends. The crucial ingredient will be the appropriate choice of the function $p_L$ in (12) for the condition on the boundary at asymptotic infinity, such that it properly reflects the feature of a codimension one system, where the gauge potential can permeate one dimension more than the induced current. To derive this condition, consider first an ordinary $(3+1)$-dimensional system. The longitudinal boundary condition can be derived rather straightforwardly from the Coulomb potential. For simplicity, we work in Coulomb gauge $A_t = \phi$ and $A_x = 0$. In the absence of external sources, a perturbation of the internal charge density $\delta\rho$ must be related to a change in the potential,

$$\delta\phi(t, \boldsymbol{r}) = \int \mathrm{d}^3 r' V(|\boldsymbol{r} - \boldsymbol{r}'|)\delta\rho(t, \boldsymbol{r}'). \tag{15}$$

Before proceeding, we emphasize that even though this seem like an instantaneous Coulomb interaction, this is the fully relativistic result following from a retarded interaction. The apparent conundrum is simply due to the fact that the choice of Coulomb gauge makes the interaction just look instantaneous, while it, of course, still preserves causality – see e.g. [18].

At any rate, after Fourier-transforming we get

$$\delta\phi(\omega, \boldsymbol{k}) = V(\boldsymbol{k})\delta\rho(\omega, \boldsymbol{k}). \tag{16}$$

With a standard Coulomb potential, $V(\boldsymbol{k}) = \boldsymbol{k}^{-2}$, in terms of bulk fields this corresponds to

$$\delta A_t - \frac{1}{k^2}\delta J_t = 0, \qquad \delta A_x = 0. \tag{17}$$

Though, when working in a holographic description at fixed chemical potential, it is more convenient to gauge transform these conditions into

$$\omega^2 \delta A_x + \delta J_x = 0, \qquad \delta A_t = 0. \tag{18}$$

This is exactly the plasmon condition (10) for codimension zero. However, if the charges are

confined to a plane $z = 0$, the integral (15) changes to

$$\delta\phi(t, \boldsymbol{r}) = \int \mathrm{d}^3 r' \delta(z) V(|\boldsymbol{r} - \boldsymbol{r}'|) \delta\rho(t, \boldsymbol{r}'). \tag{19}$$

The $\delta$-function can be incorporated into the potential to an effective potential $V_{\mathrm{eff}}$ for longitudinal modes. In Fourier-space,

$$\delta\phi(\omega, \boldsymbol{k}) = V_{\mathrm{eff}}(\boldsymbol{k}) \delta\rho(\omega, \boldsymbol{k}). \tag{20}$$

For a standard Coulomb potential $V(\boldsymbol{k}) = \boldsymbol{k}^{-2}$,

$$V_{\mathrm{eff}}(\boldsymbol{k}) = \frac{2}{|\boldsymbol{k}|}, \tag{21}$$

and this leads to the boundary conditions being adjusted with a factor $k/2$. That is, in the $(2+1)$-dimensional boundary, the boundary conditions are instead

$$\omega^2 \delta A_x + \frac{|k|}{2} \delta J_x = 0, \qquad \delta A_t = 0. \tag{22}$$

Thus, the foremost effect of restricting the currents to one dimension less than the gauge potential is a factor of $|k|/2$ in the plasmon boundary conditions. And in what follows we will demonstrate how this provides an accurate holographic model of the physics of 2D plasmons.

For illustrative purposes we will also introduce a toy model that is representative of a large class of holographic models considered in the literature and which contains all the key ingredients to demonstrate the effect of (22), while still being simple enough to not get bogged down with technicalities involving implementation that would only distract from the core issue at hand. To avoid the difficulties of the dynamics when it comes to gravitational back-reaction to the brane embedding, we will, as a proxy, just take a simple lower-dimensional charged system coupled directly to gravity, like a planar Reissner-Nordström black hole. This is sufficient because the exact details of the back-reaction are of secondary importance and have little influence on qualitative results, the main ingredient is having the proper boundary conditions to identify the collective modes. Regarding the latter, we then have to remedy the fact that we actually would have one additional spatial dimension to include. This we achieve by including the factor $|k|/2$ into the boundary condition (12), and we will show below that this is indeed the crucial ingredient to find a physically accurate response.

## 5 Results

As mentioned above, we simulate a holographic codimension one system by taking a planar Reissner-Nordström model with a $(2+1)$-dimensional boundary. As far as conventions and basic setup are concerned, we stay consistent with previous work [3], though for the sake of consistency and being self-contained, the reader will also find necessary detail and model specifics in appendix A. To counteract that test-charges in this description would, implicitly, tend to have the wrong potential, we include the corrective factor $|k|/2$ in the plasmon condition (12), which would be there if interactions in the system would, correctly, be described by electromagnetism in $(3+1)$-dimensions. The longitudinal result is presented in figures 2a and 2b. For a non-zero charge, the dispersion relation is $\omega \propto \sqrt{k}$ for small $k$, while for large $k$ the dispersion changes to $\omega \approx k$, i.e. the dispersion of a (zero) sound mode. The holographic model we have constructed therefore correctly captures the behavior of codimension one materials like a sheet of graphene. The small $k$ behavior can in fact be derived analytically to be

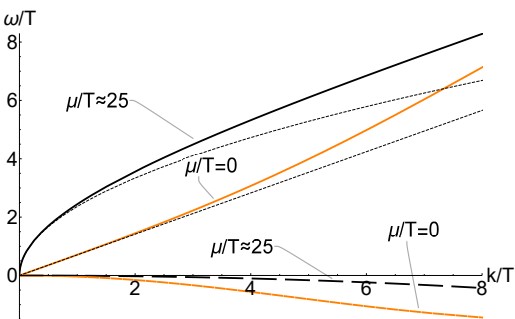

(a) The lowest collective mode for the RN background at $\mu/T = 0$ and $\mu/T \approx 25$ with $p = k/2$. Imaginary parts are negative and dashed. The thin dotted black lines are the expected $k \to 0$ asymptotes $\omega = k/\sqrt{2}$ and $\omega \propto \sqrt{k}$ respectively.

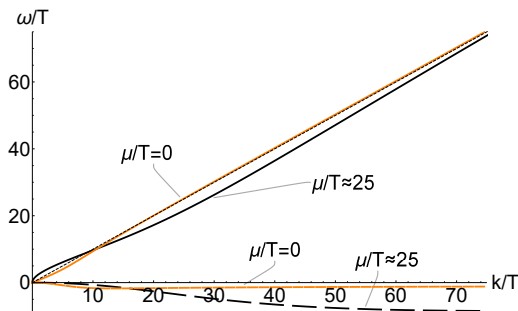

(b) The lowest collective mode for the RN background at $\mu/T = 0$ and $\mu/T \approx 25$ with $p = k/2$. Imaginary parts are negative and dashed. The thin dotted black line (mostly inside the orange line) is the expected $k \to \infty$ asymptote $\omega = k$.

Figure 2

$$\omega(k) = \sqrt{\frac{2Q^2}{6+3Q^2}} \sqrt{\lambda k} - i\left(\frac{6-Q^2}{12+6Q^2}\right)^2 \lambda k + \cdots, \tag{23}$$

where $\lambda$ is a parameter relating the bulk and boundary electromagnetic coupling constants, which do not have to be chosen to be the same, in principle. However, since the relative coupling strength $\lambda$ does not qualitatively affect the results in the model we study, we will work with $\lambda = 1$ in the following, and refer to [3] for a thorough discussion. We also wish to emphasize that while the prefactors in (23) are of course very model-dependent, having a consistent hydrodynamic expansion with $\mathcal{O}(k) = \mathcal{O}(\omega^2)$ is a direct consequence of the $|k|/2$ factor in the codimension one plasmon boundary condition (22). And thus, naturally, a low $k$ dispersion relation with $\omega \propto \sqrt{k}$ will be the generic outcome.

Furthermore, the consistency of the model can be checked by considering that it must be subject to certain sum rules. And, within numerical precision, we indeed can verify that $\lim_{k\to 0} \int_0^\infty d\omega\, \text{Im}[\,\epsilon(\omega,k)^{-1}/\omega\,] = -\pi/2$, as required, c.f. [19] for details. The apparent divergence of the (derivative of the) dispersion at $k \to 0$ is expected for charges perfectly confined to a plane. A less perfect confinement, which in an ideal model would correspond to smearing of the branes, would introduce a cut-off at small enough $k$ with an instead linear dispersion. The dispersion is thus not to be confused with that of plasmons at an *interface*, where charges are not strictly confined to a plane, but have a finite penetration depth into the material away from the conducting layer.

Beyond that, we can reach regions of parameter space which have only recently become accessible to experiments, such as $\mu \approx T$ relevant for the Dirac fluid in graphene [20–22]. The extreme case $\mu/T = 0$ is possible to observe for truly neutral media (such as He-3), and our result for this case can be seen in figures 2a and 2b. Here, we can note that for small $k$ there is a hydrodynamic (first) sound mode, $\text{Re}[\omega] \approx k/\sqrt{2}$ and $\text{Im}[\omega] \propto -k^2$, and a collisionless (zero) sound mode, $\text{Re}[\omega] \approx k$ and $\text{Im}[\omega] \propto -\text{const}$, for large k. This is in accordance with the expectation that the plasmon mode of charged systems turns into the sound mode for neutral systems [23]. Since we can access intermediate values of $\mu/T$, which is very challenging to access with standard CMT techniques, we can also predict how this happens, with the result being that there are three regions. For small $k$, there is the plasmon $\omega \propto \sqrt{k}$, for intermediate $k$, there is (first) sound, $\text{Re}[\omega] \approx k/\sqrt{2}$ and $\text{Im}[\omega] \propto -k^2$, and for large $k$, there is the collisionless (zero) sound mode, $\text{Re}[\omega] \approx k$ and $\text{Im}[\omega] \propto -\text{const}$. One way to present these different regions is by plotting the derivative of the logarithms,

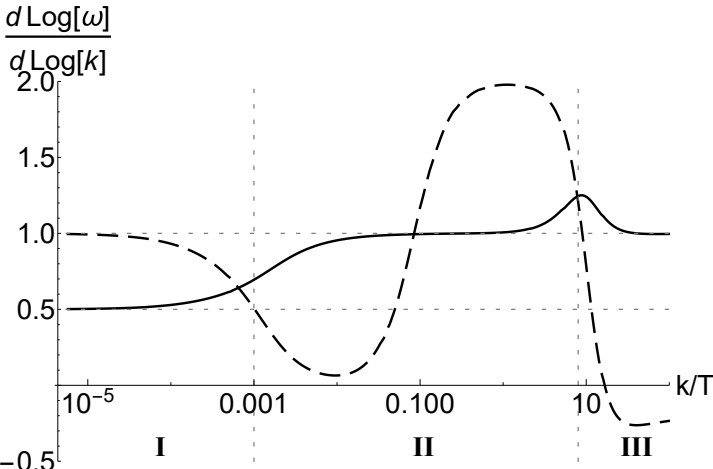

Figure 3: The derivatives $\frac{d \log \mathrm{Re}[\omega]}{d \log k}$ (solid), respectively $\frac{d \log -\mathrm{Im}[\omega]}{d \log k}$ (dashed) for the lowest collective mode at $\mu/T \approx 0.08$ with $p = k/2$. Note how $\omega \propto \sqrt{k} + i\Gamma k \to k + i\Gamma k^2 \to k + i$ const.

shown in figure 3. There is a distinction into, roughly, three different regions, which by using nomenclature from studies of dispersion relations in holographic models [24], we will denote as hydrodynamic (I), collisionless thermal (II) and collisionless quantum (III) regime.

In particular, note the significant difference in physics between the two linear sound modes, for small and large $k$ respectively, in figures 2a and 2b. Working in units where $c = 1$, the (first) sound mode for small $k$ has speed $1/\sqrt{2}$ and is increasingly unstable when increasing $k$. The (zero) sound mode for large $k$, representing collective oscillations of the Fermi surface where such exist, has speed 1 and an imaginary part largely unaffected by increasing $k$.

It is important to emphasize that the results in figure 3 are very generic for any type of model for 2D plasmons considered in 'applied holography'. The behavior in the collisionless quantum regime (III) is generic when having a relativistic theory in the bulk. And the behavior in the hydrodynamic regime (I) is a direct consequence of the boundary condition (22) for holographic plasmons in codimension one systems – respectively a consistent hydrodynamic expansion with $\mathcal{O}(k) = \mathcal{O}(\omega^2)$ induced by it. Generalizations of the toy model considered, by e.g. adding more fields, will mostly add more features in the collisionless thermal regime (II) due to the introduction of additional scales, but as long as gravitational and electromagnetic interactions in the bulk are not fundamentally altered, the qualitative behavior seen in regimes (I) and (III) will remain the same.

It is also worth stressing that the dispersion we compute holographically agrees with the results from conventional condensed matter approaches for the regions of configuration space where comparisons can be made. In particular, in the small $k$ region where $\omega \propto \sqrt{k}$, in addition to matching the real part of the dispersion also the linear dependence of the imaginary part of $\omega$ on $k$, c.f. figure 3, matches recent computations for this "collisionless plasmon" part of the configuration space, and so does the $k$-independent imaginary part of $\omega$ for large $k$ in the zero sound [23]. Furthermore, for small $k$ the results match hydrodynamic[4] results [25], but since a scattering parameter is not explicitly included in our model, it is difficult to distinguish between the hydrodynamic and "collisionless plasmon" regimes.

---

[4]For non-zero values of the intrinsic electrical conductivity $\sigma_Q$, which is the generic case for the hydrodynamic limit of holographic models. Other common names for $\sigma_Q$ are the quantum critical or incoherent conductivity.

# 6 Conclusion

In this paper we derive the conditions that all collective modes in holography must satisfy, both in the longitudinal and transverse sectors, thereby extending and further substantiating the holographic formalism pioneered in [3]. We find that for isotropic systems there are boundary conditions that translate the necessary features from the boundary theory into the bulk, (12) and (13) for longitudinal and transverse collective modes, respectively. We also point out that, in the case of holographic Maxwell theories, poles of the screened correlator $\chi_{sc}$, corresponding to QNMs, are not collective modes as they require non-vanishing external fields, and hence represent *driven excitations* of the system.

We then demonstrate that this method of relating the physical set-up in the boundary theory to the boundary conditions on the equations of motion in the bulk is not only instructive and intuitive, but also a powerful tool when regarding other types of Maxwell theories, such as codimension one materials. Such systems often show a characteristic $\omega \propto \sqrt{k}$ dispersion relation for 2D plasmons, which for graphene has been argued on general considerations involving RPA and other condensed matter methods [26, 27], and been observed in experiments [4–6]. It is therefore a very conclusive proof of consistency of the holographic plasmon formalism, introduced in [3], that this small $k$ behavior for 2D plasmons will emerge naturally from just a few very basic assumptions. A dynamically back-reacted metric and confinement to a codimension one surface being the only ingredients, we have argued that the coupling of the gravitational and electromagnetic sectors will reproduce the correct 2D plasmon dispersion relation in the hydrodynamic regime. We have further demonstrated this by simulations of an illustrative toy model, where the $\omega \propto \sqrt{k}$ dispersion can be seen to dominate over a range of several orders of magnitudes before it transitions into the collisionless thermal regime in an intermediate range, and then into the natural $\omega \sim k$ behavior in the collisionless quantum regime for large $k$. We then have also established an argumentation why this behavior is in fact generic for holographic 2D plasmons. Reproducing this behavior in a holographic model, let alone have it emerge naturally, has previously been impossible both in bottom-up systems, since the mode is heavily dimension dependent, and in top-down systems, since the mode also requires a dynamically back-reacted metric.

The use of mixed boundary conditions that are tailored to the boundary theory opens up a wide range of possible follow-up studies to previous work, as they can be applied to previously examined bulk theories. Especially interesting are models that include a physically more realistic mechanism for dynamical polarization, and back-reacted top-down models that have a bulk theory that better aligns with the boundary theory in codimension one systems while still keeping back-reacted gravity effects in the bulk.

Finally, the methods pioneered in this paper also hold the potential for making experimentally relevant predictions for 2D plasmon physics in geometries difficult to analyse using standard techniques. One such case concerns the plasmon properties in conical geometries close to the tip where field enhancement effects lead to strong interactions[5]. In addition, recent advances in experimental techniques targeting strange metals has led to the first experimental data on plasmon properties for this interesting class of materials [2]. So far there are few data points to guide attempts for holographic model building, but this is expected to change when further progress yield more data in the near future.

---

[5]We thank Bert Hecht for bringing this open problem to our attention.

# Acknowledgements

We would like to thank Johanna Erdmenger, Bert Hecht and Tobias Wenger for valuable discussions.

**Funding information**   This work is supported by the Swedish Research Council.

# A   The Planar RN Model

The planar $AdS_4$-RN model is governed by the Einstein-Maxwell action

$$S = -\int d^4x \sqrt{g}\left(-\tfrac{1}{4}F_{\mu\nu}F^{\mu\nu} + \frac{3}{L^2} + \frac{R}{2}\right) + \int_{z\to 0} d^3x \sqrt{\gamma}\left(-2K + \frac{2}{L}\right), \qquad (24)$$

where $L$ is the AdS length scale, $\gamma$ is the induced boundary metric, $K = \gamma^{\mu\nu}\nabla_\mu n_\nu$ and $n_\mu$ is the normal vector to the boundary. The dynamics is given by the first term in (24), and the standard boundary terms, i.e. the Gibbons-Hawking term and a constant term [28,29], are included to yield Dirichlet boundary conditions on the metric and make the action finite, respectively. Note that in the context of holographic renormalization no counter-term is needed for the Maxwell part of the action, since the gauge field falls off sufficiently fast when approaching the boundary in $AdS_4$, see e.g. [30]. A radial coordinate $z$, can be chosen such that the black hole horizon is located at $z = 1$ and the conformal boundary at $z = 0$. The static solution then has the metric

$$ds^2 = \frac{L^2}{z^2}\left(-f(z)dt^2 + dx^2 + dy^2 + \frac{1}{f(z)}dz^2\right), \qquad (25)$$

and a gauge field with only a time component, $A_t$. The equations of motion for this background become

$$f'(z) = \frac{z^4\left(A'_t(z)/L\right)^2 + 6f(z) - 6}{2z}, \qquad (26)$$

$$A''_t(z) = 0. \qquad (27)$$

Requiring $f$ and $A_t$ to vanish at $z = 1$ (which defines the horizon respectively eliminates a divergence) the different RN-backgrounds are determined by a single parameter $Q$, with

$$f(z) = 1 - \left(1 + \tfrac{1}{2}Q^2\right)z^3 + \tfrac{1}{2}Q^2z^4, \qquad (28)$$

$$A_t(z) = L\,Q(1-z). \qquad (29)$$

This parameter is related to the only relevant dimensionless quantity on the boundary, $\mu/T$, as

$$\frac{\mu}{T} = \frac{8\pi Q}{6 - Q^2}, \qquad (30)$$

where the boundary temperature $T$ is given by the Hawking temperature of the black hole, and the chemical potential $\mu$, can be read off with the holographic dictionary as the boundary value of $A_t$.

To study the longitudinal excitations of this system, plane wave perturbations with $(\omega, \boldsymbol{k} = k\hat{x})$ are made in the $g_{tt}$, $g_{tx}$, $g_{xx}$, $g_{yy}$, $A_t$ and $A_x$ components. These perturbations

are treated in linear response, that is,

$$g_{\mu\nu} \to g_{\mu\nu} + \epsilon \frac{L^2}{z^2} e^{-i\omega t + ikx} \delta g_{\mu\nu}, \tag{31}$$

$$A_\mu \to A_\mu + \epsilon L e^{-i\omega t + ikx} \delta A_\mu, \tag{32}$$

with some small parameter $\epsilon$. The resulting six equations of motion are shown in appendix B. There are in general 6 solutions to these equation that are relevant, two of them in-falling, which need to be solved for, or pure gauge. To be a physical excitation, there must exist a non-trivial linear combination of these solutions that satisfy the boundary conditions at $z = 0$, which is only the case for some values of $(\omega, k)$. For the metric components, the conditions are that they should vanish on the boundary. The remaining two boundary conditions are determined by the type of electromagnetic modes one wishes to describe on the boundary, e.g. (22). Computationally, finding such $(\omega, k)$ is done most conveniently by studying the determinant of the matrix of solutions and their respective boundary values,

$$\left. \begin{vmatrix} \delta g_{tt}(z)_1 & \delta g_{tx}(z)_1 & \delta g_{xx}(z)_1 & \delta g_{yy}(z)_1 & \delta A_t(z)_1 & \left(\omega^2 \delta A_x + p_L \, \delta A_x'(z)\right)_1 \\ \delta g_{tt}(z)_2 & \delta g_{tx}(z)_2 & \delta g_{xx}(z)_2 & \delta g_{yy}(z)_2 & \delta A_t(z)_2 & \left(\omega^2 \delta A_x + p_L \, \delta A_x'(z)\right)_2 \\ & & \cdots & & & \end{vmatrix} \right|_{z \to 0}, \tag{33}$$

and finding for which values of $\omega$ and $k$ it vanishes. When the determinant is zero, there must exist a linear combination of the individual solutions that satisfy all boundary conditions. Sweeping over $\omega$ and $k$ yields the dispersion relations of the boundary theory, shown in figures 2a, 2b and 3.

## B  Equations of Motion for the Perturbations

The six second order differential equations for the perturbations become as follows:
Equation for $\delta g_{tt}$:

$$\left(-\frac{f'}{2f} - \frac{2}{z}\right) \delta g_{tt}' - \frac{1}{4} f' \delta g_{xx}' - \frac{1}{4} f' \delta g_{yy}' + \frac{\delta g_{tt}\left(\left(f'\right)^2 - f\left(k^2 + Q^2 z^2\right)\right)}{2f^2}$$
$$- \frac{\delta g_{yy}\left(f k^2 + \omega^2\right)}{2f} - \frac{k\omega \delta g_{tx}}{f} - \frac{\omega^2 \delta g_{xx}}{2f} - 3Q z^2 \delta A_t' + \delta g_{tt}'' = 0.$$

Equation for $\delta g_{tx}$:

$$\frac{k\omega \delta g_{yy}}{f} - 2Q z^2 \delta A_x' + \delta g_{tx}'' - \frac{2\delta g_{tx}'}{z} = 0.$$

Equation for $\delta g_{xx}$:

$$\frac{\delta g_{tt}\left(k^2 + Q^2 z^2\right)}{2f^2} - \frac{\delta g_{yy}\left(f k^2 + \omega^2\right)}{2f^2} + \frac{k\omega \delta g_{tx}}{f^2} + \frac{\omega^2 \delta g_{xx}}{2f^2} + \left(\frac{3f'}{4f} - \frac{2}{z}\right) \delta g_{xx}' - \frac{f' \delta g_{yy}'}{4f}$$
$$- \frac{Q z^2 \delta A_t'}{f} + \delta g_{xx}'' = 0.$$

Equation for $\delta g_{yy}$:

$$-\frac{\delta g_{tt}\left(k^2 - Q^2 z^2\right)}{2f^2} + \frac{\delta g_{yy}\left(\omega^2 - f k^2\right)}{2f^2} - \frac{k\omega \delta g_{tx}}{f^2} - \frac{\omega^2 \delta g_{xx}}{2f^2} - \frac{f' \delta g_{xx}'}{4f} + \left(\frac{3f'}{4f} - \frac{2}{z}\right) \delta g_{yy}'$$
$$- \frac{Q z^2 \delta A_t'}{f} + \delta g_{yy}'' = 0.$$

Equation for $\delta A_t$:

$$\frac{Qf'\delta g_{tt}}{2f^2} - \frac{k^2\delta A_t}{f} - \frac{k\omega\delta A_x}{f} - \frac{Q\delta g'_{tt}}{2f} - \frac{Q\delta g'_{xx}}{2} - \frac{Q\delta g'_{yy}}{2} + \delta A''_t = 0 \,.$$

Equation for $\delta A_x$:

$$\frac{k\omega\delta A_t}{f^2} + \frac{\omega^2\delta A_x}{f^2} + \frac{f'\delta A'_x}{f} - \frac{Q\delta g'_{tx}}{f} + \delta A''_x = 0 \,.$$

From varying the action, an additional four first order differential equations are obtained. These are found in the $z$-components, and should automatically be satisfied by a solution to the equations above, but can be used to validate a found solution. These constraint equations are as follows:

From the $g_{tz}$-variation:

$$-\frac{kf'\delta g_{tx}}{f} - \frac{\omega f'\delta g_{xx}}{2f} - \frac{\omega f'\delta g_{yy}}{2f} + k\delta g'_{tx} + \omega\delta g'_{xx} + \omega\delta g'_{yy} = 0 \,.$$

From the $g_{xz}$-variation:

$$-\frac{kf'\delta g_{tt}}{2f} - fk\delta g'_{yy} - 2kQz^2\delta A_t + k\delta g'_{tt} - 2Q\omega z^2\delta A_x + \omega\delta g'_{tx} = 0 \,.$$

From the $g_{zz}$-variation:

$$\frac{\delta g_{tt}\left(-2f' + k^2 z + Q^2 z^3\right)}{2f} + \left(\frac{zf'}{4} - f\right)\delta g'_{xx} + \left(\frac{zf'}{4} - f\right)\delta g'_{yy} + \frac{z\delta g_{yy}\left(\omega^2 - fk^2\right)}{2f}$$
$$+ \frac{k\omega z\delta g_{tx}}{f} + \frac{\omega^2 z\delta g_{xx}}{2f} - Qz^3\delta A'_t + \delta g'_{tt} = 0 \,.$$

From the $A_z$-variation:

$$2fk\delta A'_x - \frac{Q\omega\delta g_{tt}}{f} - 2kQ\delta g_{tx} - Q\omega\delta g_{xx} - Q\omega\delta g_{yy} + 2\omega\delta A'_t = 0 \,.$$

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
