# Peer review of "Plasmons in Holographic Graphene"

_SciPost Physics, doi:SciPost Phys. 8, 093 (2020)_

## Round 4 · Referee Report · Anonymous (Referee 1) · 2020-3-27

Strengths

1) I find the question of plasmons in holography very interesting. 2) The introduction is well written and motivates the rest of the paper.

Weaknesses

1) The main weakness of the paper is the lack of clarity for the technical details on the actual computation in the bulk spacetime. 2) I would like to see aspects of holographic renormalisation to be discussed somewhere in the paper.

Report

I find the question the authors try to answer very interesting. However, at this stage, the draft seems to be unclear in terms of the technical details of the computation. To reach a satisfactory level of clarity, the authors need to state what equations they are solving in the bulk and with what boundary conditions. They only seem to kind of focus on the latter but it is not done in a very direct way.

---

## Round 5 · Referee Report · Anonymous (Referee 1) · 2020-5-31

Report

After reading the new manuscript and the response of the authors, I am happy to recommend its publication without further delays.

---

## Round 5 · Author Response

We thank the referee for his/her positive comments regarding the question addressed in the paper and for the constructive criticism.

To address the lack of clarity regarding technical details, we have extended appendix A by adding technical details regarding how we perform the linear response analysis, and in the new appendix B we have listed the equations of motion for the perturbations that we solve. We considered this linear response analysis to be part of the standard lore, but adding the details clearly makes the paper more self-contained.

We have also added a discussion regarding holographic renormalization in the beginning of appendix A, where the action is introduced, and added the two standard counterterms explicitly in the action (they were of course used in the previous computations). Note that no counterterm is necessary for the Maxwell part of the action.

We hope that after these additions, addressing the concerns of the referee, the paper will be judged ready for publication.

---

## Round 5 · List of Changes

Appendix A: Extended to include technical details regarding how we perform the linear response analysis.
Appendix A: Discussion regarding holographic renormalisation added after eq (24), where the action is introduced, and the two standard counterterms have been written out explicitly in the action.
New appendix B added containing all the equations of motion for the perturbations that we solve.

---

## Editorial Decision

published